# Piezoelectric Performance of a Symmetrical Ring-Shaped Piezoelectric Energy Harvester Using PZT-5H under a Temperature Gradient

**DOI:** 10.3390/mi11070640

**Published:** 2020-06-29

**Authors:** Nannan Zhou, Rongqi Li, Hongrui Ao, Chuanbing Zhang, Hongyuan Jiang

**Affiliations:** Department of Machine Design, School of Mechatronics Engineering, Harbin Institute of Technology, Harbin 150001, China; 19S008143@stu.hit.edu.cn (N.Z.); 19S008144@stu.hit.edu.cn (R.L.); zhangchuanbing@hit.edu.cn (C.Z.)

**Keywords:** piezoelectric energy harvester, temperature gradient, piezoelectric performance

## Abstract

With the rapid development of microelectronics technology, low-power electronic sensors have been widely applied in many fields, such as Internet of Things, aerospace, and so on. In this paper, a symmetrical ring-shaped piezoelectric energy harvester (SR-PEH) is designed to provide energy for the sensor to detect the ambient temperature. The finite element method is used by utilizing software COMSOL 5.4, and the electromechanical coupling model of the piezoelectric cantilever is established. The output performance equations are proposed; the microelectromechanical system (MEMS) integration process of the SR-PEH, circuit, and sensor is stated; and the changing trend of the output power density is explained from an energy perspective. In the logarithmic coordinate system, the results indicate that the output voltage and output power are approximately linear with the temperature when the resistance is constant. In addition, the growth rate of the output voltage and output power decreases with an increase of resistance under the condition of constant temperature. In addition, with an increase of temperature, the growth rate of the output power is faster than that of the output voltage. Furthermore, resistance has a more dramatic effect on the output voltage, whereas temperature has a more significant effect on the output power. More importantly, the comparison with the conventional cantilever-shaped piezoelectric energy harvester (CC-PEH) shows that the SR-PEH can improve the output performance and broaden the frequency band.

## 1. Introduction

With the progress of microelectronics technology, wireless sensor technology has practical applications in environmental monitoring, aerospace, and Internet of Things [1,2,3]. Wireless sensors can complete data collection by analyzing electrical signals and transferring information to the gateway through radio frequency (RF) modules. However, the working life of wireless sensors is usually limited by energy supply components. Traditional chemical batteries no longer meet the requirements of sensors due to limitations such as limited life and environmental pollution [4]. Micro/nano technology, as well as mechanical and material engineering technology make the collection and application of vibration energy possible [5,6]. Vibration energy is mainly converted into electrical energy through electromagnetic conversion, electrostatic conversion, piezoelectric conversion, magnetostrictive conversion, photoelectric conversion, and other methods [7,8,9,10,11]. A comparison with other forms of energy harvesters shows that the piezoelectric energy harvester has the advantages of simple structure, no heat generation, higher output power, and no pollution to the environment [12].

Researchers have proposed various structures to improve the piezoelectric efficiency of piezoelectric energy harvesters. The cantilever has a low resonance frequency and is easy to integrate with a microelectromechanical system (MEMS), and therefore it is a focus issue in the energy harvester field [13,14]. Z.J. Li proposed the embedded connection method of the piezoelectric cantilever [15]. In his research, the resonance frequency of the embedded cantilever was reduced by 43.1% as compared with a conventional piezoelectric cantilever. Additionally, the output voltage and output power of the embedded cantilever were increased by 2.83 times and 7.3 times, respectively. S. Srinivasulu Raju connected the rectangular cantilever and the tapered cantilever with a tapered ratio *β* in forward and reversed directions [16]. The output voltage of the connected cantilever was increased by 91.3% and 76.9%, respectively, as compared with the rectangular cantilever. Saxena designed four cantilever-type MEMS-based piezoelectric energy harvesters to power sensors that detected the health of a building [17]. The results illustrated potential generated ranges from 17.82 mW to 89.21 mW for input acceleration ranging from 1 g to 5 g, respectively. B. Abdul explored the output performance of the segmented cantilever under different conditions [18]. The results indicated that the output voltage and output power of a segmented cantilever were higher than that of a conventional cantilever. P.H. Wang simulated and experimentally analyzed the array rectangular piezoelectric cantilever [19] and verified that the array piezoelectric cantilever could effectively increase the working frequency band. L. Jin investigated the output performance of a MEMS piezoelectric energy harvester under different width shapes [20]. In the research, the output performance of the harvester was tested by setting different cantilever structures. M.J. Huang developed a MEMS piezoelectric energy harvester system that achieved frequency conversion [21]. The power density of this system reached 0.216 µW·g^−1^·Hz^−1^·cm^−3^ at 0.3 g acceleration. In order to comprehensively consider the advantages of rectangular cantilevers, segmented cantilevers, and array structures, in this paper, we propose a symmetrical ring-shaped piezoelectric energy harvester (SR-PEH) based on previous research results which can supply energy for a wireless sensor to detect ambient temperature.

PZT-5H has the advantages of excellent piezoelectric output performance as compared with PVDF and AlN [22,23,24,25]. PZT-5H combines the advantages of bulk materials and film materials, which allows it to work in low-voltage and high-frequency environments. In this study, we selected PZT-5H as the piezoelectric material to improve the output performance of the harvester.

In the present study, we explore the output performance of a SR-PEH and analyze the influencing factors of the output performance. The equations illustrate the changing characteristics and influencing parameters of the output performance. This research compares a conventional cantilever-shaped piezoelectric energy harvester (CC-PEH) with a SR-PEH and concludes that the SR-PEH has better output performance. This research has value as a particular reference for the design of a high-performance piezoelectric energy harvester.

## 2. Structure and Method

### 2.1. Structure Design

The working method of piezoelectric material is direct piezoelectricity [26]. Polarized charges are generated on the upper and lower surfaces of the piezoelectric material under the influence of stress and strain [27]. There are usually two modes for piezoelectric materials to generate polarized charges, as shown in Figure 1. The *d*_31_ mode indicates that the stress direction is perpendicular to the electric field direction, whereas the *d*_33_ mode expresses that the stress direction is parallel to the electric field direction [28]. A cantilever-type piezoelectric energy harvester generally adopts the *d*_31_ mode.

The structure of piezoelectric energy harvester has a significant influence on the output performance. Figure 2 shows the structure of the SR-PEH and the components of the piezoelectric sheets. The specific structural sizes and physical parameters are shown in Table 1.

MEMS is an advanced intelligent system, which is also widely used in different fields. In the design of MEMS, the use of top-down design methods can effectively complete the integrated design, and also can achieve the mutual decomposition of the design links, which ensures the rationality of the MEMS, and also allows the MEMS to have more features. A SR-PEH, as the energy supply element of a sensor, can be integrated with an energy collection circuit and sensor using the MEMS integrated method, as shown in Figure 3.

### 2.2. Finite Element Method (FEM)

The finite element method (FEM, software COMSOL Multiphysics 5.4) was utilized to analyze the piezoelectric performance of the SR-PEH. The modal analysis illustrated that the first-order resonance frequency of the SR-PEH was 140 Hz. In the simulation process, the fixed end of the cantilever was fully constrained, and the simulation conditions were 100~200 Hz frequency domain and 1 g acceleration. The physical modules included the solid mechanics, the solid heat transfer, the electrostatic, and the circuit. In this study, a 4-node free tetrahedron element was used for mesh generation. In addition, the grid of piezoelectric materials was refined to improve calculation accuracy. The influences of the conductive adhesive layer, insulation layer, and electrode on the simulation results were neglected during the solution process to improve the computational efficiency. Figure 4 exhibits the meshing and first-order mode of vibration.

FEM uses the stress-charge and strain-charge piezoelectric equations to achieve the conversion of force-electricity calculations. The calculation efficiency of stress-charge is higher than that of strain-charge, therefore, for this research, we selected the stress-charge constitutive piezoelectric equation.

The linear electrical property of piezoelectric material is expressed in Equation (1).
(1)D=εE
where *D* is electric displacement vector, *ε* expresses relative permittivity, and *E* denotes electric field, and
(2)∇∙D=0,∇×E=0

Hooke’s law is satisfied when piezoelectric materials are in the linear region, as shown in Equation (3).
(3)δ=sσ
where *δ* is strain, *s* expresses compliance under short-circuit conditions, and *σ* expresses stress. Besides,
(4)$$\nabla ∙\sigma =0,\delta =\frac{1}{2}\left(\nabla u+\nabla u^{T}\right)$$

Therefore, the stress-charge constitutive piezoelectric equation of piezoelectric material is obtained in Equation (5).
(5)$$\left\{ \begin{array}{l} \sigma =c\delta -eE\\ D=e\delta +\varepsilon E \end{array}\right.$$
where *c* denotes elastic stiffness matrix, *e* expresses piezoelectric stress constant matrix.

### 2.3. Electromechanical Coupling Model

Piezoelectric materials produce polarized charges under the effect of stress and strain, which realizes the conversion of vibration energy into electrical energy. The relationship between the deformation of the piezoelectric cantilever and output power is derived in this section, and the electromechanical coupling model of the piezoelectric cantilever is shown in Figure 5. In order to simplify the calculation, this deduction assumes that the length and width of the substrate are equal to those of PZT-5H. The *h*_T_ and *h*_H_ are the thickness of the substrate and PZT-5H, respectively. *M* represents the mass, *V* expresses the output voltage, and *z*(x) denotes the dynamic deflection equation.

The relationship between PZT-5H stress and strain is shown in Equation (6).
(6)$$\left\{ \begin{array}{l} T_{H}=E_{H}(S_{1}-g_{31}D_{3})\\ e_{3}=-g_{31}T_{H}+\beta_{33}^{T}D_{3} \end{array}\right.$$
where *T*_H_ is *x*-direction stress, *E*_H_ is elastic modulus of PZT-5H, *S*_1_
*= −rz* denotes *x*-direction strain, *g*_31_ expresses piezoelectric voltage constant, *D*_3_ represents electrical displacement in the *z*-direction, and *e*_3_ denotes electric field strength in the *z*-direction. $\beta_{33}^{T}$ indicates dielectric isolation rate, $\beta_{33}^{T}=1/\varepsilon_{33}^{T}$, $\varepsilon_{33}^{T}$ expresses dielectric constant in *z*-direction.

The neutral layer position of the piezoelectric cantilever is calculated according to Equation (7).
(7)$$z_{N}=\frac{E_{H}\frac{h_{H}{}^{2}}{2}+E_{T}h_{T}(h_{H}+\frac{h_{T}}{2})}{E_{H}h_{H}+E_{T}h_{T}}$$
where *E*_T_ denotes elastic modulus of the substrate.

The equivalent stiffness is solved by the static bending equation of cantilever in Equations (8) and (9).
(8)$$E_{\mathrm{TH}}I_{\mathrm{TH}}=E_{H}\int_{0}^{b}\int_{-h_{H}}^{z_{N}}zdzdy+E_{T}\int_{0}^{b}\int_{z_{N}}^{h_{T}}zdzdy$$
(9)$$I_{\mathrm{TH}}=\int_{0}^{b}\int_{-h_{H}}^{h_{T}}zdzdy$$
where *I*_TH_ is extreme moment of inertia of the piezoelectric cantilever to the *x*-axis, and *b* is piezoelectric cantilever width.

The curvature equation of the piezoelectric cantilever is shown in Equation (10).
(10)$$\frac{1}{r_{\mathrm{TH}}(x)}=\frac{\partial^{2}z_{\mathrm{TH}}(x)}{\partial x^{2}}$$
where *z*_TH_ is the dynamic deflection equation of piezoelectric cantilever, and *r*_TH_ is the curvature radius of the piezoelectric cantilever at any position.

The vibration function of the piezoelectric cantilever is expressed in Equations (11) and (12).
(11)$$\frac{\partial^{4}Z_{\mathrm{TH}}(x)}{\partial x^{4}}=-\frac{\rho_{e}A}{E_{\mathrm{TH}}I_{\mathrm{TH}}}\frac{\partial^{2}Z_{\mathrm{TH}}(x)}{\partial x^{2}}$$
(12)$$\frac{1}{r_{\mathrm{TH}}}=\frac{d^{2}Z_{\mathrm{TH}}(x)}{dx^{2}}$$
where *Z*_TH_(*x*) indicates vibration function, $\rho_{e}$ expresses equivalent density of piezoelectric cantilever, and A is the cross-sectional area of the piezoelectric cantilever.

Combining Equations (5)~(12), the electric field distribution *e*_3_ can be obtained, and its equation is shown in Equation (13).
(13)$$e_{3}=-g_{31}E_{H}(r_{\mathrm{TH}}(x)z+g_{31}D_{3})+\beta_{33}^{T}D_{3}$$

The output voltage *V* is expressed in Equation (14).
(14)$$V=\int_{-h_{H}}^{z_{N}}e_{3}dz=-g_{31}E_{H}∙\frac{z_{N}^{2}-h_{H}^{2}}{2}∙r_{\mathrm{TH}}(x)+(z_{N}+h_{H})(g_{31}^{2}E_{H}D_{3}+\beta_{33}^{T}D_{3})$$

*Q*_TH_ is the generalized charge and is shown by Equation (15).
(15)$$Q_{\mathrm{TH}}=\int_{0}^{b}\int_{0}^{1}D_{3}dydz=C_{e}V_{0}$$

The open-circuit voltage *V*_0_ is obtained by Equations (14) and (15), as follows in Equation (16).
(16)$$V_{0}=\frac{bl-2(z_{N}+h_{H})(g_{31}^{2}E_{H}+\beta_{33}^{T})C_{e}}{g_{31}E_{H}(z_{N}^{2}-h_{H}^{2})r_{\mathrm{TH}}(x)}$$
where *C*_e_ is the equivalent capacitance of PZT-5H.

The output power *P* is expressed by Equation (17).
(17)$$P=\frac{{\left[bl-2(z_{N}+h_{H})(g_{31}^{2}E_{H}+\beta_{33}^{T})C_{e}\right]}^{2}R}{{\left[g_{31}E_{H}(z_{N}^{2}-h_{H}^{2})r_{\mathrm{TH}}(x)\right]}^{2}{\left(R+R_{m}\right)}^{2}}$$
where *l* denotes the length of the PZT-5H, *R*_m_ expresses the internal resistance of the PZT-5H, and *R* is external resistance.

### 2.4. Commutating and Voltage-Stabilizing Circuit

Efficient energy recovery circuit can reduce energy consumption and achieve high-efficiency storage of electrical energy [29]. The standard energy recovery circuit and charge synchronous extraction circuit have considerable limitations. In particular, the collection efficiency is only 50% when the output voltage and output power are low. This section utilizes an improved commutating and voltage-stabilizing circuit to improve collection efficiency and reduce energy loss during the work process. The principle is shown in Figure 6.

The circuit is a significant proportion of the harvester. In order to minimize the volume of the SR-PEH, we need to integrate the circuit with the sensor. In the present research, all the circuit components and wires are integrated on the silicon substrate by etching, and then they are packaged in a shell.

The contact pins, P2 and P3, are applied to connect the PZT-5H and circuit. The electrical signals generated by the SR-PEH realize rectification through their respective rectifier bridges (DB107). The circuit realizes the low-pass filtering through the capacitor *C*1, and capacitor *C*2 serves to store charge temporarily. MAX6433 is a voltage detection chip that monitors the capacitor voltage status in real time. The “on” voltage and “off” voltage of the entire circuit can be controlled through the external resistance. The component parameters are as follows.
(18)$$V_{\mathrm{REF}}=615\mathrm{mV}$$
(19)$$V_{\mathrm{TRIPHIGH}}=V_{\mathrm{HTH}}=V_{\mathrm{REF}}\frac{R_{1}+R_{2}+R_{3}}{R_{3}}$$
(20)$$V_{\mathrm{TRIPLOW}}=V_{\mathrm{LTH}}=V_{\mathrm{REF}}\frac{R_{1}+R_{2}+R_{3}}{R_{2}+R_{3}}$$

*V*_REF_ is the reference voltage inside MAX6433. The circuit turns on after a short delay when the voltage reaches the turn-on threshold *V*_HTH_. The circuit turns off immediately when the voltage is lower than the turn-off threshold *V*_LTH_. The function of MAX666 is to achieve stable voltage output, and the assignment of LED1 is to check whether the circuit is a pathway.

## 3. Result and Discussion

### 3.1. Voltage Analysis

FEM indicates that the output voltage of the SR-PEH has a linear region and a nonlinear region when PZT-5H is affected by resistance, as shown in Figure 7a. The PZT-5H can improve life and ensure the long-term stable supply of electrical energy when it is working in the linear region. Figure 7a shows that the output voltage and resistance express a linear relationship to a certain extent when the resistance is in the range of 10~60 kΩ at 20 °C.

Figure 7b shows the output voltage of the SR-PEH at a temperature gradient of 30~100 °C. The results illustrate that the logarithm of the output voltage is approximately linear with the temperature when the resistance is constant. In addition, the logarithm of the output voltage increases with an increase of the resistance when the temperature is constant, but the growth rate gradually decreases. The equation is required by fitting the data of the output voltage under a 30~100 °C temperature gradient.
(21)$$\ln V=7.7\times 10^{-3}(T-30)+\ln R+0.077$$

Equation (21) indicates the relationship among voltage, resistance, and temperature of SR-PEH. It can be used to calculate the output voltage of the SR-PEH at a specific temperature, as shown in Figure 8.

Figure 8 shows that the resistance has a more considerable influence on the output voltage than temperature under the 30~100 °C temperature gradient. One possible reason is that the amount of polarized charge reaches saturation at a specific temperature, and higher temperatures can no longer improve the output performance of the SR-PEH. PZT-5H is the same as magnetic material, its piezoelectric effect is related to temperature. The piezoelectric performance of PZT-5H disappears if the temperature exceeds Curie temperature (150 °C). Therefore, the operating temperature of PZT-5H should be lower than its Curie temperature to ensure the output performance of the SR-PEH. This paper shows the output performance of the SR-PEH under a temperature gradient of 30~100 °C, which includes the temperature variation range of most environments. The output performance of the SR-PEH above 150 °C is not discussed in this paper.

### 3.2. Power Analysis

Under the condition that the resistance is 10~60 kΩ, FEM illustrates that the output power and the resistance are approximately linear when the temperature is 20 °C, as shown in Figure 9a. The simulation parameters of output power are the same as those of exploring the output voltage. Figure 9b shows the changing trend of output power under a temperature gradient.

Figure 9b indicates that there is an approximately linear relationship between the logarithm of output power and temperature when the resistance is constant. Under certain temperature conditions, as the resistance increases, the logarithm of the output power also increases. However, the logarithm of the output power growth rate gradually decreases. The equation of output power is obtained as follows:(22)$$\ln P=1.52\times 10^{-2}(T-30)+\ln R-1.338$$

Equations (21) and (22) illustrate that as the temperature rises, the logarithm of the output power grows faster than that of the output voltage, as shown in Figure 10.

Figure 10 shows that the difference between the output power and the output voltage decreases as the temperature increases, which indicates that the influence of temperature on the output power is more significant than that on the output voltage.

### 3.3. Power Density and Energy Analysis

The power density reflects the output power per unit volume of the SR-PEH. High power density expresses excellent output performance. Figure 11 shows the output power density and atomic energy under a temperature gradient.

Figure 11a shows that as the temperature increases, the growth rate of power density also rises when the resistance is constant. Figure 11b expresses that the atomic energy increases linearly when the temperature continues to rise. In addition, the amount of charge in the piezoelectric material increases when the temperature rises [30]. Therefore, the growth rate of power density shows an upward trend.

### 3.4. Output Performance Comparison

This section compares the output performance of the SR-PEH and CC-PEH under the same simulation parameters to illustrate the advantages of the SR-PEH, as shown in Figure 12. Frequency band and output performance are the key indicators of the energy harvester. The fuller the frequency band, the higher the piezoelectric conversion efficiency of the piezoelectric energy harvester.

Figure 12 shows that the output performance of the SR-PEH is nearly 15% higher than that of the CC-PEH when the temperature is 20 °C and resistance is 10 kΩ. In addition, when the output voltage is above 5 V, the frequency band of the SR-PEH is 3 Hz more than that of the CC-PEH. The output power shows the same phenomenon as the output voltage, which shows that the SR-PEH has a broadband function.

## 4. Conclusions

The piezoelectric energy harvester has considerable advantages for improving energy harvester life and preventing environmental pollution. In this paper, the MEMS integration process of the SR-PEH, circuit, and sensor was described, and the electromechanical coupling model of the piezoelectric cantilever was established.

(1) The output performance of the SR-PEH was explored by using the FEM under a temperature gradient of 30~100 °C and 1 g acceleration. The results demonstrate that the logarithm of output voltage and output power is approximately linear with temperature when the resistance is constant. The logarithm of output voltage and output power gradually increases with an increase of resistance when the temperature is constant, but their growth rate gradually decreases. Compared with temperature, resistance has more influence on output power.

(2) The equations of piezoelectric performance were proposed to illustrate the relationship between temperature, resistance, output performance. The equation illustrates that the growth rate of the logarithm of output power is higher than that of the logarithm of the output voltage. In addition, the change of output power is more dramatic than that of output voltage under the condition of temperature variations.

(3) The reason for the change in power density was explained from the perspective of atomic energy. The results show that the increase in temperature leads to the rise of atomic energy and charge quantity, and therefore the growth rate of power density also increases.

(4) The output performances of the SR-PEH and the CC-PEH were compared to illustrate the excellent piezoelectric performance of the SR-PEH. The results express that the SR-PEH has the capacity to improve output performance and broadband as compared with the CC-PEH.

Piezoelectric energy harvesters are anticipated to be the right choice for powering wireless sensors. The SR-PEH proposed in this paper has value as a particular reference for the design of piezoelectric energy harvesters.

## Figures and Tables

**Figure 1 micromachines-11-00640-f001:**
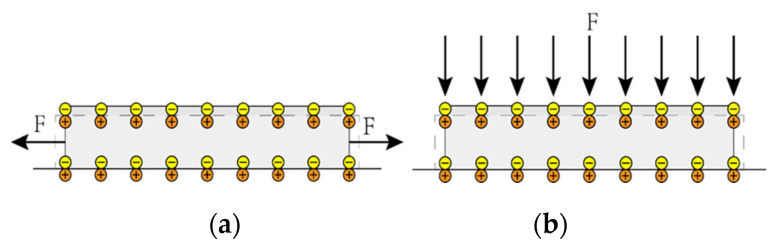
Two working modes. (**a**) *d*_31_ mode; (**b**) *d*_33_ mode.

**Figure 2 micromachines-11-00640-f002:**
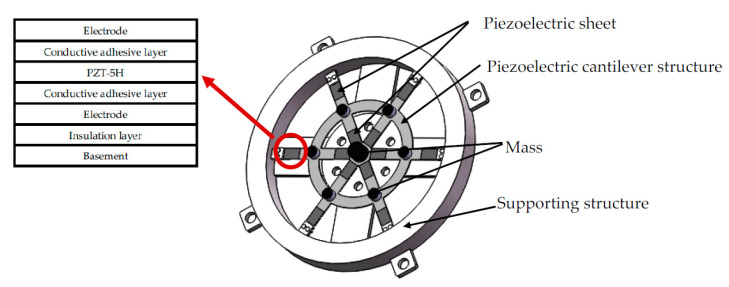
The symmetrical ring-shaped piezoelectric energy harvester (SR-PEH).

**Figure 3 micromachines-11-00640-f003:**
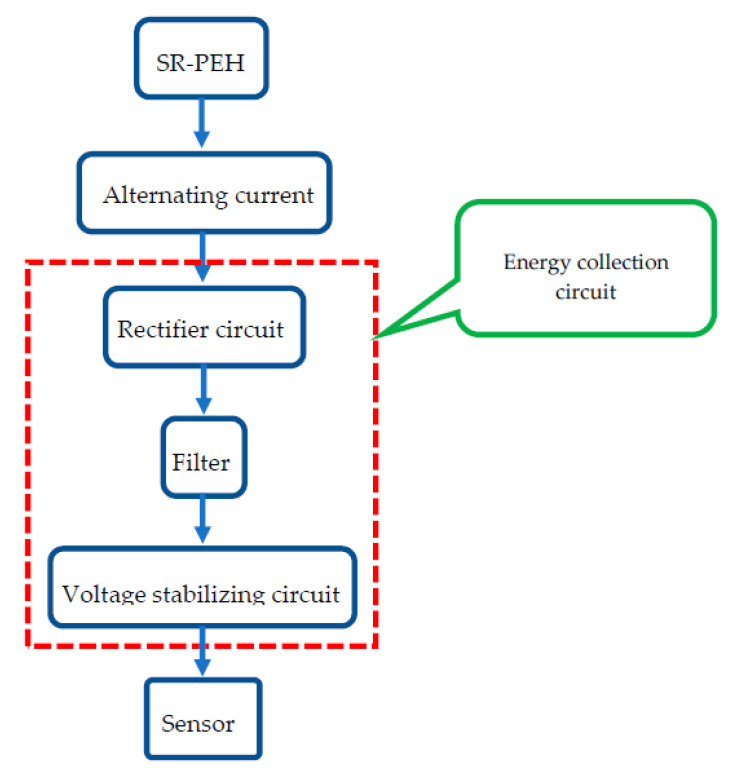
The microelectromechanical system (MEMS) integration process.

**Figure 4 micromachines-11-00640-f004:**
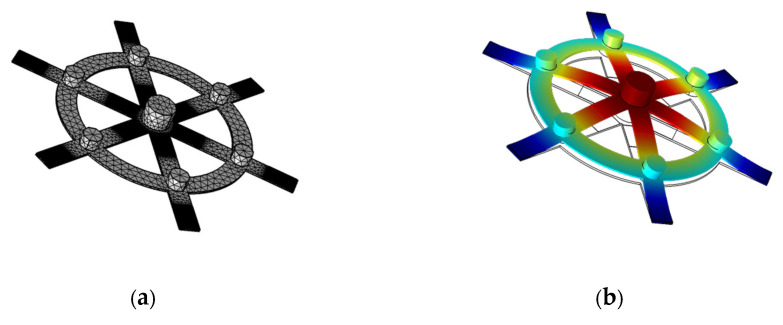
The finite element method (FME) results of the SR-PEH. (**a**) Meshing; (**b**) First-order mode of vibration.

**Figure 5 micromachines-11-00640-f005:**
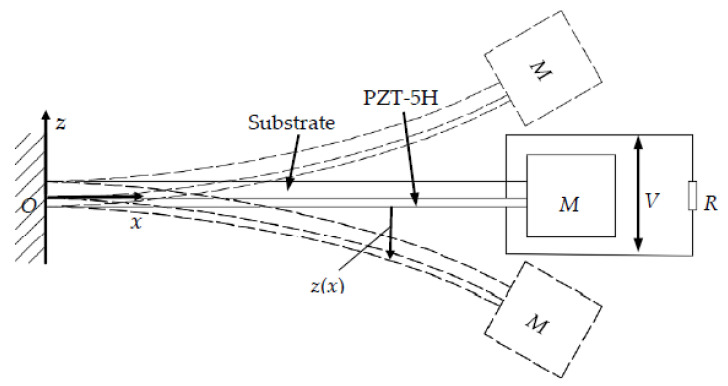
Electromechanical coupling model.

**Figure 6 micromachines-11-00640-f006:**
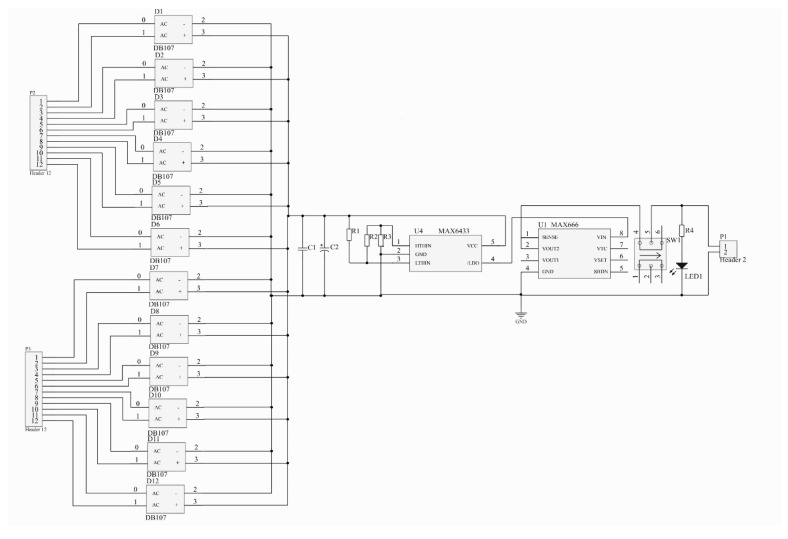
Commutating and voltage-stabilizing circuit.

**Figure 7 micromachines-11-00640-f007:**
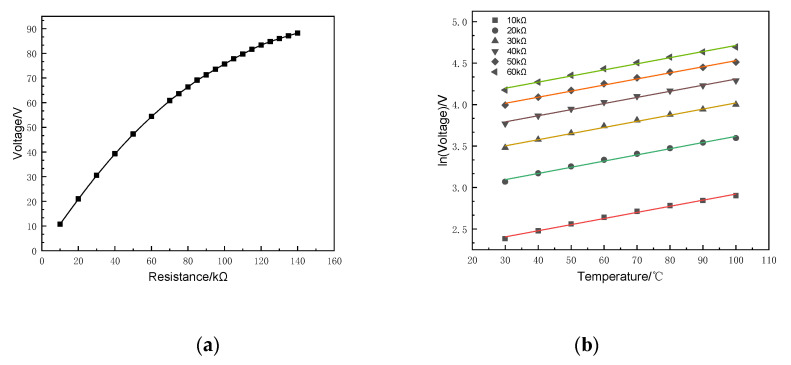
Output voltage at different temperatures. (**a**) At 20 °C; (**b**) At temperature gradient.

**Figure 8 micromachines-11-00640-f008:**
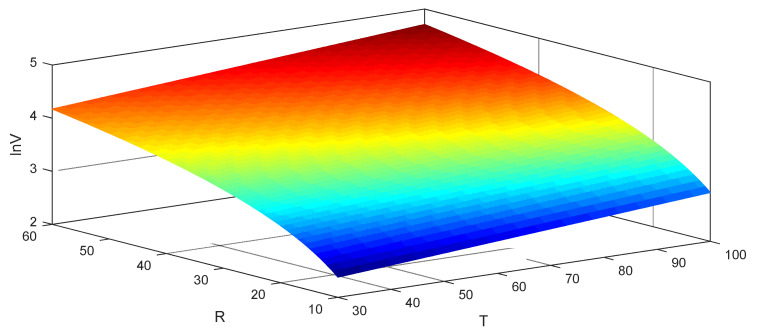
Output voltage under continuous parameters.

**Figure 9 micromachines-11-00640-f009:**
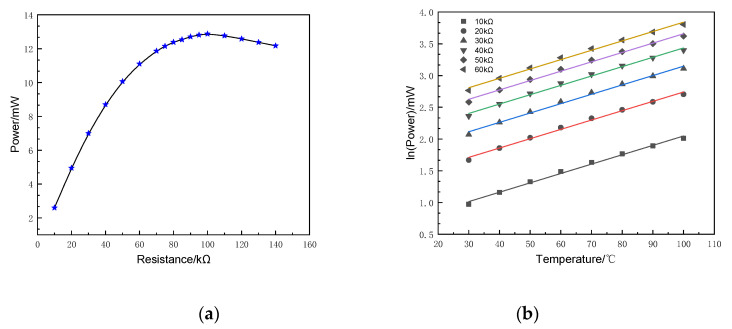
Output power at different temperatures. (**a**) At 20 °C; (**b**) At temperature gradient.

**Figure 10 micromachines-11-00640-f010:**
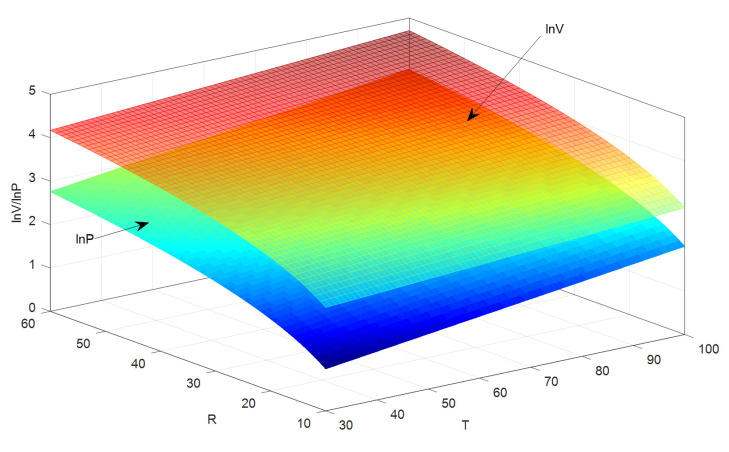
Comparison of the output voltage and the output power.

**Figure 11 micromachines-11-00640-f011:**
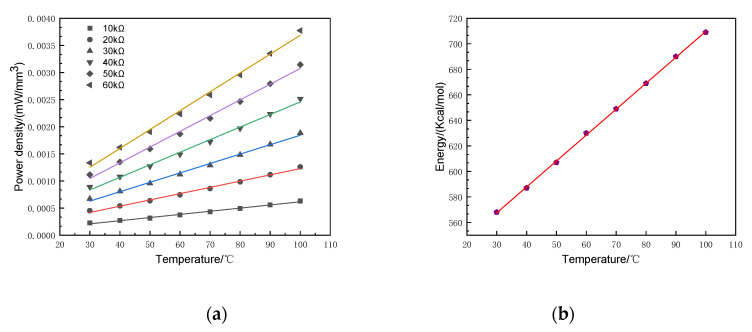
Power density and energy analysis. (**a**) Power density; (**b**) Atomic energy.

**Figure 12 micromachines-11-00640-f012:**
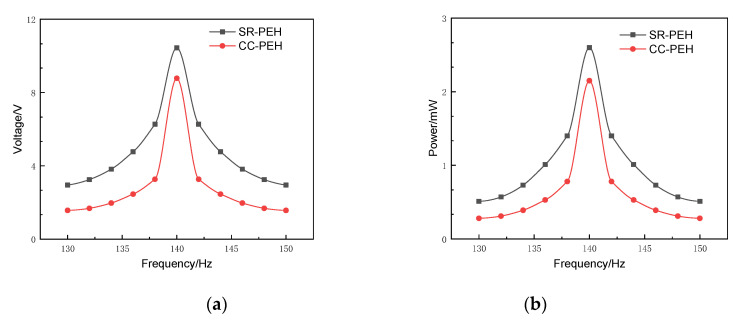
Output performance comparison. (**a**) Output voltage comparison; (**b**) Output power comparison.

**Table 1 micromachines-11-00640-t001:** Structural sizes and physical parameters.

Heading	Material	Size (mm)	Piezoelectric Coefficient *d*_31_	Resistivity
Piezoelectric material	PZT-5H	15 × 10 × 0.15	−41~274	High
Substrate	Copper	150 × 10 × 1	-	Low
Insulation layer	PMMA	16 × 10 × 10^−4^	-	High
Conductive adhesive layer	Epoxy resin	16 × 10 × 10^−4^	-	Low
Electrode	Chromium	15 × 10 × 10^−4^	-	Low
Supporting structure	PMMA	-	-	High

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
