# Peer review of "Piezoelectric Performance of a Symmetrical Ring-Shaped Piezoelectric Energy Harvester Using PZT-5H under a Temperature Gradient"

_micromachines, 2020, doi:10.3390/mi11070640_

Round 1

Reviewer 1 Report

This research designs a symmetrical ring-shaped piezoelectric energy harvester (SR-PEH) using the finite element analysis by COMSOL Software. It can provide an effective way and structure for the important component in the field of piezoelectric energy harvesting and/or MEMS devices. This will be helpful to various researchers for developing efficient piezoelectric systems. It can be published after some minor revisions.

  1. This research is a theoretical study using the finite element method. However, the authors show a figure (Fig. 3) about the processing. Why do the authors present it? Moreover, the figure is too detailed, which is not related to this research (e.g., Cu substrate, Cr sputtering, etc.). The process is not absolute in the field. I think it is unnecessary in this manuscript.
  2. The manuscript should cite more recent references about MEMS to attract broader readership as follows.
  • Nano Energy, 2019, 55, 182-192.
  • Nano Energy, 2019, 55, 211-219.

Author Response

Dear reviewer:

General comments: This research designs a symmetrical ring-shaped piezoelectric energy harvester (SR-PEH) using the finite element analysis by COMSOL Software. It can provide an effective way and structure for the important component in the field of piezoelectric energy harvesting and/or MEMS devices. This will be helpful to various researchers for developing efficient piezoelectric systems. It can be published after some minor revisions.

Response: We thank the reviewer for the positive comments and the important suggestions. We provide our response to the comments by the reviewer below:

Question1: This research is a theoretical study using the finite element method. However, the authors show a figure (Fig. 3) about the processing. Why do the authors present it? Moreover, the figure is too detailed, which is not related to this research (e.g., Cu substrate, Cr sputtering, etc.). The process is not absolute in the field. I think it is unnecessary in this manuscript.

Response: According to your suggestion, we have removed the content about the process in the paper. Because the process is not absolute in piezoelelctric field. The symmetrical ring-shaped piezoelectric energy harvester (SR-PEH), as the energy supply element of sensor, needs to integrate with energy collection circuit and sensor to achieve its function. Figure 3 shows the MEMS integration process of SR-PEH, circuit, and sensor, which may increase the integrity and coherence of the paper.

Note: Figure 3 is shown in the attachment below.

Question 2:The manuscript should cite more recent references about MEMS to attract broader readership as follows.

Response: In this paper, the following three papers about MEMS piezoelectric energy harvester are added to make readers understand some practical applications of MEMS technology in piezoelectric field.

1. Saxena, R. Sharma, B.D.Pant. Design and development of guided four beam cantilever type MEMS based piezoelectric energy harvester. Microsystem technologies. 2017, 23, 1751-1759.

2. L Jin, S Q Gao, X Y Zhang. Output of MEMS Piezoelectric energy harvester of Double-Clamped Beams With Different Width Shapes. Materials. 2020, 13(10), 2330.

3. M J Huang, C Hou, Y F Li. A Low-Frequency Piezoelectric Energy Harvesting System Based on Frequency Up-Conversion Mechanism. Micromachines. 2019, 10(10), 639.

Reviewer 2 Report

I would not call the harvested energy "massive."  It is mW.

Figure 8 with Temperature and Resistance and output voltage seem to indicate the high temperature (100) and high Resistance produces more voltage.  IS that accurate?  Do you get close to the Curie TEmperature? The material you picked has a Curie of 150 - just add a discussion.

Author Response

Dear reviewer:

We thank the reviewer for the positive comments and the important suggestions. We provide our response to the comments by the reviewer below:

Question1:I would not call the harvested energy "massive”. It is mW.

Response: Compared with other forms of energy harvester, such as electrostatic conversion, magnetostrictive conversion, photoelectric conversion, the piezoelectric energy harvester has the advantage of higher output power, but only mW. Therefore, this paper uses “higher” to replace “massive”, which not only highlights the advantages of piezoelectric energy harvester, but also does not cause readers to misunderstand.

Question2:Figure 8 with Temperature and Resistance and output voltage seem to indicate the high temperature (100) and high Resistance produces more voltage. IS that accurate? Do you get close to the Curie temperature? The material you picked has a Curie of 150 - just add a discussion.

Response:Like magnetic materials, the piezoelectric effect of PZT-5H is related to temperature. The piezoelectric performance of PZT-5H will disappear if the temperature exceeds Curie temperature (150℃). Therefore, the operating temperature of PZT-5H should be lower than its Curie temperature to ensure the output performance of symmetrical ring-shaped piezoelectric energy harvester (SR-PEH). Under normal circumstances, the temperature of the environment is far below 100 ℃, for example, the maximum temperature of the equator will not exceed 50 ℃, the extreme temperature of hot springs on the earth is lower than 75 ℃. This paper shows that the output performance of SR-PEH under a temperature gradient of 30℃~100℃, which includes the temperature variation range of most environments, as shown in Figure 8. The output performance of SR-PEH above 150 ℃ is not discussed in this paper.

Note:Figure 8 is shown in the attachment below.
